# From Theory to Practice: Implementing the WHO 2021 Classification of Adult Diffuse Gliomas in Neuropathology Diagnosis

**DOI:** 10.3390/brainsci13050817

**Published:** 2023-05-18

**Authors:** Karina Chornenka Martin, Crystal Ma, Stephen Yip

**Affiliations:** 1Department of Pathology & Laboratory Medicine, Faculty of Medicine, University of British Columbia, Vancouver, BC V5Z 1M9, Canada; karina.chornenka@vch.ca; 2Faculty of Medicine, University of British Columbia, Vancouver, BC V6T 2A1, Canada; cma7338@student.ubc.ca

**Keywords:** diffuse glioma, astrocytoma, IDH-mutant, oligodendroglioma, 1p19q-codeleted, glioblastoma, IDH-wildtype, integrative diagnosis

## Abstract

Diffuse gliomas are the most common type of primary central nervous system (CNS) neoplasm to affect the adult population. The diagnosis of adult diffuse gliomas is dependent upon the integration of morphological features of the tumour with its underlying molecular alterations, and the integrative diagnosis has become of increased importance in the fifth edition of the WHO classification of CNS neoplasms (WHO CNS5). The three major diagnostic entities of adult diffuse gliomas are as follows: (1) astrocytoma, IDH-mutant; (2) oligodendroglioma, IDH-mutant and 1p/19q-codeleted; and (3) glioblastoma, IDH-wildtype. The aim of this review is to summarize the pathophysiology, pathology, molecular characteristics, and major diagnostic updates encountered in WHO CNS5 of adult diffuse gliomas. Finally, the application of implementing the necessary molecular tests for diagnostic workup of these entities in the pathology laboratory setting is discussed.

## 1. Introduction

Primary central nervous system (CNS) neoplasms are the eighth most common malignancy, but represent the highest mortality rate amongst human cancers. Diffuse gliomas are the most frequent type of primary brain malignancy to affect the adult population [1], and their diagnosis is becoming increasingly reliant on the integration of the histomorphology of these tumours with their molecular features [2]. Adult diffuse gliomas are a group of neoplasms that demonstrate infiltrative growth, with tumour cells characteristically percolating through normal CNS cellular components, making these entities incredibly challenging to treat. Maximal safe surgical resection is an important component of treatment, but alone, it is insufficient to irradicate disease. Patients often require post-operative adjuvant chemoradiation therapy; however, local recurrences and distant tumour progression occur in the majority of cases. Certain cases with hypermutated phenotypes characterized by loss of mismatch repair proteins compound another layer of therapeutic difficulty as these tumours demonstrate resistance to conventional treatment modalities [3,4].

The three major diagnostic entities of adult diffuse gliomas are as follows: (1) astrocytoma, IDH-mutant; (2) oligodendroglioma, IDH-mutant and 1p/19q-codeleted; and (3) glioblastoma, IDH-wildtype. The aim of this review is to outline the pathology, molecular characteristics, and major diagnostic updates of adult diffuse gliomas within the fifth edition of the WHO classification of CNS neoplasms (WHO CNS5) [5] and discuss the application of the necessary diagnostic workup in the laboratory setting [6].

## 2. Astrocytoma, IDH-Mutant

IDH-mutant astrocytomas have an average annual incidence of 1.21 per 100,000, and typically occur in individuals younger than 55 years of age, with a mean age of onset of 38 [1,7,8]. These tumours can occur in any region of the brain or spinal cord but are typically supratentorial with a predilection for the frontal lobes [9,10]. Infratentorial IDH-mutant astrocytomas are less common, however, they have been found to frequently possess non-canonical IDH mutations and usually lack ATRX mutations, suggesting they may represent a unique subset of IDH-mutant astrocytomas [11,12,13]. Median overall survival for patients diagnosed with IDH-mutant astrocytoma ranges between 7 and 10 years, depending on various factors such as patient age, tumour location, tumour grade, and the specific treatment protocol undertaken [1,14,15,16]. On imaging, IDH-mutant astrocytomas can be recognized with the identification of T2-FLAIR mismatch, which has been shown to be a specific imaging marker of the IDH-mutant, 1p-19q non-codeleted glioma [17,18]. With contrast enhanced imaging studies, lower grade tumours tend not to enhance, while higher grade tumours more frequently show enhancement [19,20].

### 2.1. Pathophysiology

Several studies suggest that IDH-mutant astrocytomas may originate from precursor cells of the central nervous system as they share commonalities with neural precursor cells and/or precursor cells of oligodendroglial or astrocytic lineages [21,22,23]. Additionally, these tumours are defined by mutations in either *IDH1* or *IDH2*, which is an early alteration in tumourigenesis [24]. *IDH1* and *IDH2* are homologous genes that code for isocitrate-dehydrogenase, an enzyme in the Krebs cycle that converts isocitrate to α-ketoglutarate while reducing NADP to NADPH and releasing carbon dioxide. Neomorphic mutations in *IDH1/2* result in the enzyme overproducing 2-hydroxyglutarate, an oncometabolite that plays a role in altering histone methylation patterns and promoting hypermethylation of DNA. Ultimately, this results in the glioma-associated CpG island methylator phenotype (G-CIMP), with subsequent repression of genes involved in differentiation. This promotes a neural stem cell-like state with regenerative and proliferative potential that plays a role in tumorigenesis [25,26,27]. The most common mutations in diffuse astrocytomas are missense mutations in *IDH1*, which account for over 70% of cases, with a substitution of histidine for arginine at codon 132 (IDH1 R132H) being the most frequent alteration. The most common missense mutation in *IDH2* is a substitution of lysine for arginine at codon 172 (IDH2 R172K), which represents approximately 2% of IDH-mutant astrocytomas. Other non-canonical mutations in *IDH1* or *IDH2* that occur at codons 132 and 172, respectively, are rare and together account for approximately 15% of IDH-mutant astrocytomas [28,29,30].

### 2.2. Histology and Diagnostic Workup

IDH-mutant astrocytomas are characterized by diffuse infiltration of neoplastic astrocytic tumour cells into surrounding normal brain parenchyma within a fibrillary neuropil background. As such, resection specimens show hypercellular tissue with an indistinct boundary between the tumour and background brain parenchyma. In certain cases, a gradient of infiltration may be appreciated and entrapped neurons are apparent within the infiltrating tumour (Figure 1A).

The individual tumour cells have elongated hyperchromatic nuclei and typically scant inconspicuous cytoplasm; however, they may have large bellies of glassy eosinophilic cytoplasm in gemistocytic variants (Figure 1E). Nuclear pleomorphism is often observed. In higher grade tumours, increased mitotic activity, microvascular proliferation, and/or necrosis are seen. Microvascular proliferation is determined by the presence of vessels composed of at least a bilayer of plump endothelial cells, which sometimes form glomeruloid structures. Pseudopalisading necrosis is the pattern of necrosis usually seen in high grade tumours (Figure 1F).

Immunohistochemical workup includes staining with IDH1 R132H, ATRX, p53, and Ki67. IDH-mutant astrocytomas are strongly associated with mutations in *ATRX* and *TP53* [31]. Consequently, the expected immunophenotype would show cytoplasmic immunoreactivity for IDH1 R132H, loss of nuclear ATRX immunoreactivity, and strong overexpression of p53 indicating a missense mutation (Figure 1B–D). In cases that are not immunoreactive for IDH1 R132H, but ATRX nuclear expression is lost, and the p53 staining pattern is indicative of a mutation, a non-canonical *IDH1* or *IDH2* mutation should be suspected, and sequencing should be pursued, particularly if the patient is younger than 55 years of age and/or the tumour has a lower grade histology [8]. The Ki67 proliferation index is variably elevated and may correlate with the grade of the tumour; however, it is not established as a reliable grading criterion.

### 2.3. Grading and Molecular Integration for Diagnosis and Prognosis

IDH-mutant astrocytomas are designated a grade of 2, 3, or 4 based on the presence of certain histological features and/or molecular alterations. Grade 2 tumours show diffuse infiltration of atypical cells in background brain parenchyma with a mild-to-moderate increase in cellularity compared to a normal brain. Mitotic activity is uncommon, and microvascular proliferation and necrosis are absent. Grade 3 tumours show a more densely cellular pattern with increased nuclear pleomorphism; however, the presence of increased mitoses is the definitive criterion that upgrades a grade 2 neoplasm to a grade 3 neoplasm. If microvascular proliferation or necrosis are present, this designates the tumour as grade 4.

A newly established grading criterion is introduced in WHO CNS5, which incorporates molecular features of these tumours into the final integrated diagnosis. Homozygous deletion of *CDKN2A* and/or *CDKN2B (CDKN2A/B)* defines a grade 4 assignment irrespective of the histological features. This addition to the grading criteria is based on the results of several studies showing that *CDKN2A/B* homozygous deletion results in patient outcomes similar to grade 4 tumours, even in the absence of high grade histological features [32,33,34,35]. The diagnostic approach to an IDH-mutant astrocytoma when presented with an adult diffusely infiltrating glioma is summarized in Figure 2. Another major change made in WHO CNS5 is that IDH-mutant astrocytomas that show grade 4 features are no longer termed “glioblastoma”, as this is now a diagnosis reserved for IDH-wildtype tumours (see below). This adjustment in terminology stems from growing evidence that IDH-mutant (grade 4) astrocytomas and IDH-wildtype glioblastomas clinically behave differently, with IDH-mutant tumours predicting a more favourable prognosis, and are characterized by distinct molecular alterations and epigenetic profiles [7,36,37].

Another molecular feature of these tumours that is relevant for prognosis is the presence of *MGMT* promoter methylation. *MGMT* codes for O-6-methylguanine DNA methyltransferase, a DNA repair protein that diminishes the effects of alkylating agents used to treat these gliomas, such as temozolomide and lomustine. MGMT functions by removing alkyl groups from the O6 position of guanine, thus weakening the therapeutic benefit of these agents [38,39]. The majority of IDH-mutant astrocytomas possess methylated *MGMT* promoters, and compared to IDH-wildtype glioblastomas, IDH-mutant tumours show significantly higher *MGMT* promoter methylation. Amongst IDH-mutant astrocytomas, *MGMT* promoter methylation is correlated with prolonged overall survival [40,41].

## 3. Oligodendroglioma, IDH-Mutant and 1p/19q-Codeleted

Oligodendrogliomas, IDH-mutant, 1p19q-codeleted have an average annual incidence of 0.48 per 100,000, and typically appear in patients between their fourth and fifth decade of life, with a median overall survival of 10–17 years depending on patient demographics, tumour size and location, and the pursued treatment protocol [1,42]. These tumours preferentially localize to the cerebral hemispheres, most commonly to the frontal lobes, followed by temporal and parietal locations, with occipital localization being rare [1,43,44,45]. On CT without contrast, oligodendrogliomas usually appear hypodense, and approximately 90% present with calcifications. On FLAIR MRI, the heterogeneous and infiltrative quality of the tumour into the cortex can be appreciated. Oligodendrogliomas may show mild-to-moderate heterogeneous enhancement on contrast-enhanced MRI, which is considered to be the radiologic correlate to the “chicken-wire” vasculature commonly seen in histology [46,47].

### 3.1. Pathophysiology

The pathogenesis of oligodendrogliomas is still an area of active research. Some studies have proposed that oligodendrogliomas are tumours that arise from oligodendrocyte precursor cells [48,49], while other studies have suggested that these neoplasms can originate from astrocytes and neural progenitors [50,51]. The defining molecular alterations of this tumour are missense mutations of *IDH1* or *IDH2*, in addition to the deletion of the whole arm of chromosomes 1p and 19q, with mutations in *IDH1/2* thought to be an early alteration that drives tumourigenesis prior to the establishment of 1p19q co-deletion [24]. Similar to IDH-mutant astrocytomas, IDH1 R132H represents the most common mutation in oligodendrogliomas, although non-canonical mutations in *IDH2* at codon 172 are more common in oligodendrogliomas compared to astrocytomas [28]. An unbalanced translocation between chromosomes 1 and 19, and the ensuing loss of the derivative chromosome, gives rise to the 1p19q co-deleted molecularly defining feature of this neoplasm, which also confers improved outcomes in patients with oligodendrogliomas [52,53]. Other alterations characteristic of oligodendrogliomas are *TERT* promoter and *CIC*, *FUBP1*, and *NOTCH1* mutations [7,54,55,56,57].

### 3.2. Histology and Diagnostic Workup

Oligodendrogliomas are diffuse neoplasms that send tumour cells infiltrating into the brain parenchyma. Individual tumour cells are characterized by relatively monomorphic ovoid-to-round nuclei containing stippled chromatin surrounded by perinuclear haloes, giving rise to the described “fried-egg” appearance. There is often a delicate capillary background in a characteristic “chicken-wire” pattern (Figure 3A). A subset of cases can show a minigemistocytic phenotype in which the eosinophilic cytoplasm of these cells stains strongly positive with GFAP [58]. Additionally, secondary structures are often associated with these neoplasms, such as perivascular accumulation, aggregates in the subpial space, and perineuronal satellitosis. Areas showing microcalcification, hemorrhage, and cystic degeneration with myxoid material can also be observed.

Overall, diagnosis for oligodendrogliomas is made based on the combination of histological patterns and immunohistochemical/molecular features. Immunohistochemical workup of most oligodendrogliomas shows immunoreactivity for IDH1 R132H and retention of normal nuclear ATRX (Figure 3B,C). As *TP53* mutations and 1p19q codeletions are mutually exclusive, oligodendrogliomas show a wild-type pattern of p53 staining without diffuse overexpression [7]. This immunophenotype should prompt investigation for 1p19q codeletion, which can be identified using fluorescence in situ hybridization (FISH) (Figure 3F), microarrays that survey the genome-wide copy number landscape(such as copy number microarray or the Illumina 850K methylation assay), assays utilizing microsatellite markers, next generation sequencing, and chromogenic in situ hybridization [59,60,61]. If IDH1 R132H is not immunoreactive in the setting of ATRX nuclear expression, this is likely indicative of a non-canonical *IDH1* or *IDH2* mutation and sequencing should be conducted to evaluate for this possibility.

### 3.3. Grading and Molecular Integration for Diagnosis and Prognosis

Oligodendrogliomas can be assigned a grade of 2 or 3, with the differentiating histological features being the presence of mitotic activity, microvascular proliferation, and necrosis (Figure 3D,E). In general, there is decreased overall and progression-free survival in tumours showing increased mitotic activity, microvascular proliferation, and necrosis [62,63]. These are designated as grade 3 tumours; however, specific cut-off markers for mitotic count are not yet established. Similar to IDH-mutant astrocytomas, homozygous deletion of *CDKN2A/B* in oligodendrogliomas is linked to inferior clinical outcomes, even in cases without microvascular proliferation and/or necrosis, and WHO CNS5 suggests that this may be used as a molecular marker of grade 3 tumours in cases that have borderline histology [34]. The diagnostic pathway to the histologic and molecular integration in classifying an IDH-mutant and 1p19q-codeleted oligodendroglioma is summarized in Figure 2.

## 4. Glioblastoma, IDH-Wildtype

Glioblastomas comprise the majority of malignant CNS neoplasms with an average annual incidence rate of 6.97 per 100,000. Despite being the most common malignant CNS neoplasm in the adult population, patient outcome is dismal with median overall survival ranging between 14 and 20 months, which is largely dependent on the *MGMT* promoter methylation status [64,65,66]. Though glioblastomas can affect individuals of any age, they are less common in individuals younger than 55, with a median age of onset of 65, and a peak incidence between 75 and 84 years of age [1,67]. Tumours typically localize to the subcortical white matter in any of the cerebral lobes, and a subset of glioblastomas can present as multifocal or multicentric lesions [68,69]. On contrast-enhanced MRI studies, these tumours more often show rim enhancement with central necrosis compared to other glioma subgroups [18,70].

### 4.1. Pathophysiology

The definite pathogenesis of glioblastoma is yet to be determined; however, studies suggest that glioblastomas may arise from various CNS progenitor cells, including those from neuronal, astrocytic, and oligodendroglial lineages. Specifically, various studies suggest that glioblastomas may arise from neural progenitor cells originating in the subventricular zones [71,72]. Similar to other glial neoplasms, glioblastomas are highly infiltrative, and are particularly characterized by infiltration along white matter tracts. An example of a subtype of glioblastoma that demonstrates this feature is the interhemispheric infiltration of a glioblastoma across the corpus callosum giving rise to the described “butterfly glioma” [73]. Although the majority of glioblastomas present as a solitary tumour, up to a third of patients can present with multifocal lesions. The molecular landscape of multifocal glioblastomas is similar to their solitary counterparts; however, *EGFR* mutations, and concomitant *EGFR, PTEN,* and *TERT* promoter mutations occur at an increased incidence [68,74]. Moreover, multifocal glioblastomas have a strong association with *TERT* promoter mutations in general [75], and one study suggests that 50% of these tumours are associated with c-Met overexpression based on immunohistochemistry, which is negatively correlated with survival and treatment response [76].

### 4.2. Histology and Diagnostic Workup

Glioblastomas are a heterogeneous group of tumours with many histological variants. The conventional glioblastoma is a densely cellular and infiltrative tumour characterized by fibrillary astrocytic cells with highly hyperchromatic and pleomorphic nuclei. Mitotic activity is brisk and areas of necrosis and microvascular proliferation are readily encountered (Figure 4A,B). Histologic variants of glioblastomas include the following: a giant cell subtype characterized by bizarre multinucleated giant cells, a sarcomatous subtype that is termed “gliosarcoma” (Figure 4D), an epithelioid subtype with abundant eosinophilic cytoplasm and distinct cell borders (Figure 4E), a granular cell subtype characterized by a PAS-positive granular cytoplasm, and a small cell type that can resemble a high-grade oligodendroglioma [77,78,79]. Other possible histological features include glioblastomas possessing a primitive neuronal component, epithelial metaplasia, or adipocytic differentiation [80,81,82,83].

The distinction between epithelioid glioblastoma and epithelial metaplasia is that the latter demonstrates cohesive growth, which may show squamous or adenomatous differentiation, with a subset demonstrating a true epithelial immunophenotype. On the other hand, epithelioid glioblastoma is characterized by tumour cells with ample eosinophilic cytoplasm and well-delineated cell borders, and these neoplasms may show rhabdoid differentiation. Finally, epithelioid glioblastoma show tumour cells that are generally loosely cohesive and regions may show discohesive growth [81,84,85].

Conventional glioblastomas are immunoreactive for GFAP, however, there is an inconsistent staining pattern of GFAP and, sometimes, a complete loss of expression in several of the histological subtypes is observed [79,80,86]. By definition, glioblastomas are IDH-wildtype and H3-wildtype tumours, thus immunoreactivity for IDH1 R132H and H3 p.K28M (K27M) is negative. ATRX nuclear expression is retained in the majority of glioblastomas. *TP53* alterations have been reported to occur in 22% of glioblastomas in general, but in up to 75% of giant cell glioblastomas and these cases result in overexpression in p53 on immunohistochemistry [87,88,89]. The Ki67 proliferation index is often markedly elevated in these tumours. *BRAF* V600E mutations are uncommon in glioblastomas, with the exception of epithelioid variants, which may show immunoreactivity in up to 50% of cases (Figure 4F) [90]. Although *EGFR* amplification is common in IDH-wildtype glioblastomas (Figure 4C), immunohistochemistry for EGFR has poor specificity for detecting this molecular alteration [91].

### 4.3. Grading and Molecular Integration for Diagnosis and Prognosis

IDH-wildtype glioblastomas are grade 4 tumours that are histologically characterized by high-grade features, such as microvascular proliferation and necrosis. A major change in WHO CNS5 is the designation of IDH-wildtype and H3-wildtype astrocytic tumours as glioblastomas, even in the presence of low-grade histologic features if certain key molecular criteria are met. These characteristic molecular alterations include *EGFR* amplification, combined whole chromosome 7 gain and whole chromosome 10 loss (+7/−10), and *TERT* promoter mutations. IDH-wildtype gliomas that do not fulfill these diagnostic criteria must be evaluated further to determine their appropriate WHO classifications. For example, workup may be pursued to assess for the possibility of other IDH-wildtype tumours, such as diffuse hemispheric gliomas characterized by H3 G34-mutations, diffuse midline gliomas defined by H3 K27 alterations, or MAPK pathway altered gliomas, such as pleomorphic xanthroastrocytoma (Figure 2). The integration of these molecular data into the diagnosis is supported by studies showing that histologically low-grade (i.e., grades 2 and 3) IDH-wildtype astrocytic tumours impart a substantially shorter patient survival than their IDH-mutant counterparts [7,36,37,92]. In fact, patients with histologically “low-grade” IDH-wildtype tumours that harbour *EGFR* amplification, +7/−10, or *TERT* promoter mutations demonstrate similar patient outcomes to those patients with IDH-wildtype tumours with high-grade histologic features [93,94,95].

Out of the three molecular alterations, *EGFR* amplification and +7/−10 are most specific for IDH-wildtype glioblastoma, and are strong surrogate markers for the diagnosis of glioblastoma in the absence of grade 4 histologic features [96]. *TERT* promoter mutations are the most frequent molecular alterations seen in IDH-wildtype glioblastoma, and while some studies argue that *TERT* promoter mutations are sufficient for the diagnosis of glioblastoma and predict poor prognosis [75,97,98], other studies suggest they have poor specificity as they can be encountered at an equal or greater frequency in oligodendroglioma and melanoma, and may not add additional prognostic information when other histological or molecular diagnostic criteria are met for IDH-wildtype glioblastoma [96,99,100]. Other prognostic molecular markers include homozygous deletion of *CDKN2A*/B, which imparts a poor prognosis, and critically, *MGMT* promoter methylation, which is a favourable prognostic marker showing greater overall and progression-free survival with improved response to alkylating agents and radiotherapy [101,102].

## 5. Standard of Care Molecular Testing of Adult Diffuse Glioma: Application in Neuropathology Practice

The integrative diagnosis, which involves combining disease-defining molecular features alongside descriptive morphological findings, is now a common practice in pathology and is especially advocated in WHO CNS5. The incorporation of molecular biomarkers eliminates the problems associated with histological ambiguity, interobserver variations, and the typical non-informativeness of treatment response in the era of precision oncology. However, this places additional pressure on the pathology laboratory to adopt new biomarkers and molecular tests. The latter might require extra validation costs as well as significant cost for deployment, particularly in brain tumour pathology, where the volume is less robust compared to other cancer types.

The decision for local testing versus referral of cases to a high volume outside center or commercial service is complex and can be contingent on multiple variables. However, the availability of immunohistochemical reagents to identify proteomic products of common, diagnostically relevant mutations in neuro-oncology has been a tremendous boon to the practice (Table 1). Some of these antibodies are also used in other tumour types, thus spreading the deployment cost. Similarly, technology platforms, techniques, and assays for molecular diagnostics of non-CNS malignancies can be adopted to address the detection of specific alterations in gliomas. For example, a FISH assay for the detection of *CDKN2A* homozygous deletion in mesothelioma can be adapted for use in glioma. However, if no assay exists on the test menu of an institution despite the availability of a technology platform, validation and deployment costs can be substantial, depending on the complexity of the assay. Examples include the introduction of a next generation sequencing assay to detect fusion drivers or the EPIC methylation microarray platform, which requires a standard number of validation samples, often quoted as 59, to achieve a degree of accuracy for clinical certification [103].

The cost of local assay development or remote testing at a high-volume centre can be staggering. Integration of simple in-house immunohistochemical tests and coupling the result with patient demographics and radiological features will permit the triaging of a subset of cases for more advanced molecular testing, either locally or remotely. This will help to offset the cost of uninformed testing and facilitate the speedy return of testing results.

## 6. Conclusions

Diffuse gliomas are the most common primary CNS malignancy to affect the adult population, and the three entities that constitute adult diffuse gliomas include the following: (1) astrocytoma, IDH-mutant; (2) oligodendroglioma, IDH-mutant and 1p/19q-codeleted; and (3) glioblastoma, IDH-wildtype. Each of these entities are diffusely infiltrating neoplasms, resulting in them being challenging to treat, with tumour recurrence being a common phenomenon. Gliomas that harbour an *IDH1/2* mutation (IDH-mutant astrocytoma and IDH-mutant and 1p19q-codeleted oligodendroglioma) result in improved patient outcomes as compared to the IDH-wildtype glioblastoma. *CDKN2A/B* homozygous deletion confers a worse prognosis in all three entities, while *MGMT* promoter methylation is an important favourable prognostic marker that is associated with improved overall and progression-free survival with better response to alkylating agents.

Although the histopathology varies between the three adult diffuse gliomas and can guide the neuropathologist’s initial evaluation, since the release of CNS WHO5, the landscape of adult diffuse glioma diagnosis has shifted to more heavily integrating molecular alterations that are entity and grade-defining. Immunohistochemical assessment of these neoplasms with available reagents that target protein products of key molecular alterations (Table 1) are an essential tool in neuropathology practice. Further diagnostic molecular workup, such as next generation sequencing, FISH, copy number microarray, and/or methylation profiling may be required in a subset of cases. The use of key in-house laboratory immunohistochemical tests and correlating the results with patient demographics, clinical course, and radiologic findings helps triage cases for further molecular studies, and may assist laboratories in offsetting associated costs in implementing in-house molecular assays or in referrals for remote testing.

## Figures and Tables

**Figure 1 brainsci-13-00817-f001:**
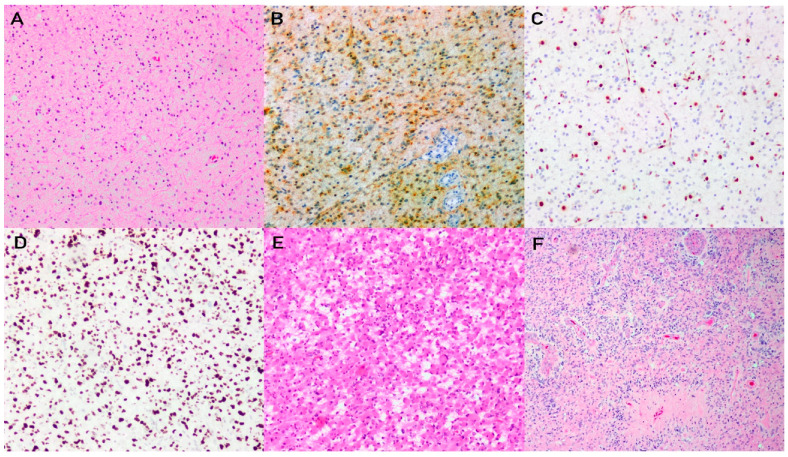
Astrocytoma, IDH-mutant. H&E stained section demonstrates a diffusely infiltrating astrocytic neoplasm (**A**). Immunohistochemistry shows IDH1 R132H cytoplasmic immunoreactivity (**B**), loss of nuclear expression of ATRX in tumour cells (**C**), and abundant staining for p53 in majority of tumour nuclei, suggestive of a *TP53* missense mutation (**D**). H&E stained section shows a diffusely infiltrating astrocytic neoplasm characterized by abundant glassy eosinophilic cytoplasm and eccentric nuclei, consistent with a gemistocytic variant of diffuse astrocytoma (**E**). H&E stained section showing a grade 4 IDH-mutant astrocytoma showing microvascular proliferation and pseudopalisading necrosis (**F**).

**Figure 2 brainsci-13-00817-f002:**
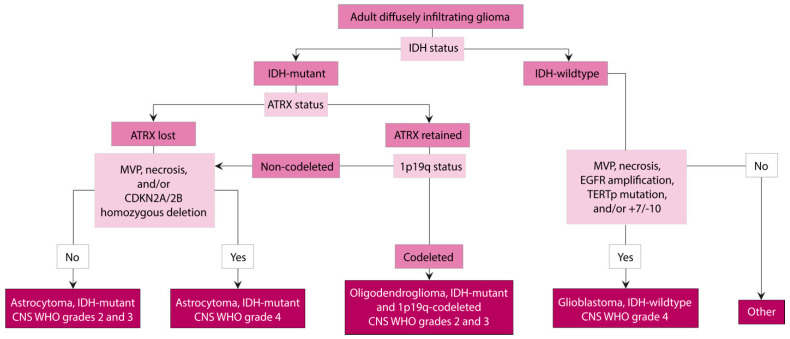
Diagnostic algorithm of adult diffusely infiltrating gliomas with integration of morphologic and molecular characteristics. Once a diffusely infiltrating glioma is identified on H&E stained sections, the tumour must be assessed for the presence of an *IDH1/2* mutation that can be detected either by immunohistochemistry for its IDH1 R132H status or with sequencing. In IDH-mutant gliomas, ATRX status is assessed with immunohistochemistry. If nuclear expression of ATRX is lost in tumour cells, this is consistent with astrocytoma, IDH-mutant. Additionally, grade 4 tumours are identified by the presence of microvascular proliferation (MVP), necrosis, and/or *CDKN2A/2B* homozygous deletion. If ATRX nuclear expression is retained, an assessment of 1p19q status should be pursued. If whole arms of chromosomes 1p and 19q are codeleted, the diagnosis is oligodendroglioma, IDH-mutant and 1p19q-codeleted. In non-codeleted tumours, the final diagnosis is astrocytoma, IDH-mutant. In IDH-wildtype diffusely infiltrating gliomas, the presence of MVP, necrosis, *EGFR* amplification, *TERT* promoter (TERTp) mutation, and/or combined whole chromosome 7 gain and whole chromosome 10 loss (+7/−10) defines a glioblastoma, IDH-wildtype. IDH-wildtype gliomas that do not fulfill these diagnostic criteria should prompt investigation for other possible entities, such as H3-mutant or MAPK driven gliomas.

**Figure 3 brainsci-13-00817-f003:**
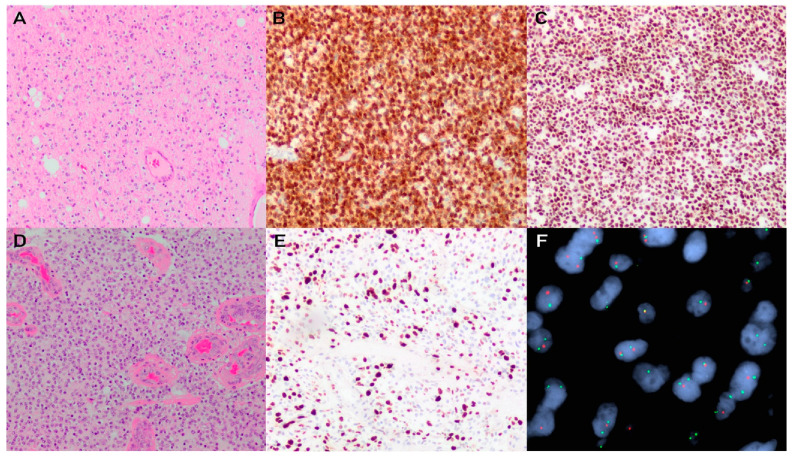
Oligodendroglioma, IDH-mutant and 1p19q-codeleted. H&E stained section showing a diffusely infiltrating glioma characterized by cells with monomorphic round nuclei and perinuclear halos giving rise to the “fried egg” appearance (**A**). Immunohistochemistry shows IDH1 R132H immunoreactivity (**B**), with retention of nuclear ATRX expression (**C**). H&E stained sections showing a more densely cellular oligodendroglial neoplasm with microvascular proliferation (**D**) and an elevated Ki67 proliferation index (**E**). Fluorescence in situ hybridization confirms 1p19q-codeletion (representative image of assessment of chromosome 1 with analogous signals achieved for chromosome 19) (**F**). There is a relative loss of the red 1p probe in comparison to the control green 1q probe in individual tumour cells.

**Figure 4 brainsci-13-00817-f004:**
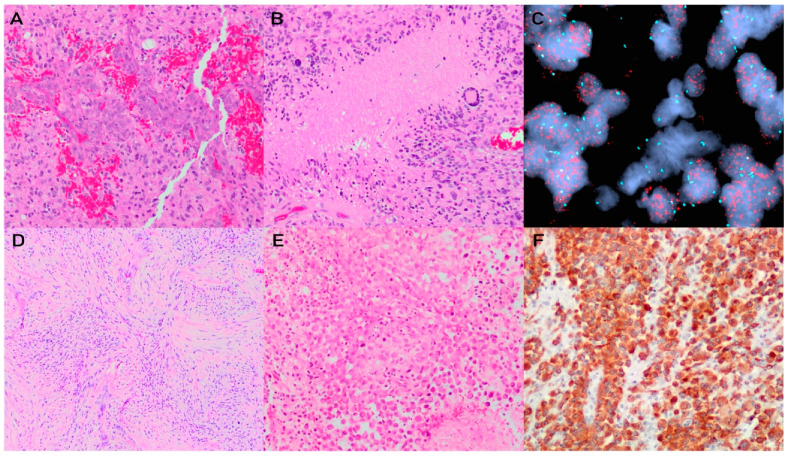
Glioblastoma, IDH-wildtype. H&E stained sections demonstrate a densely cellular and pleomorphic astrocytic neoplasm with prominent microvascular proliferation (**A**) and pseudopalisading necrosis (**B**). Fluorescence in situ hybridization reveals EGFR amplification with a significantly increased signal of the EGFR probe (red) compared to the control centromere 7 probe (green) (**C**). H&E stained sections show histologic variants of glioblastomas, including gliosarcoma (**D**) and epithelioid glioblastoma (**E**), the latter of which frequently harbours *BRAF* V600E mutations that can be detected by immunohistochemistry (**F**).

**Table 1 brainsci-13-00817-t001:** Essential glioma diagnostic tools in a clinical neuropathology practice.

Antigenic Target	Signal	Diagnostic Implications	Caveats
**IDH1 p.R132H**	Cytoplasmic POSITIVITY in tumour expressing the mutant IDH1 R132H epitope	The tumour harbours the most common IDH mutation stratifying it into the IDH-mutant astrocytoma vs. IDH-mutant oligodendroglioma diagnostic pathway.	A small percentage of IDH mutant gliomas are immunonegative secondary to non-R132H mutations in *IDH1* or mutations affecting *IDH2* requiring a secondary genetic assay for verification.
**ATRX**	Nuclear POSITIVITY in tumour cells is associated with no genomic alteration of *ATRX*	In the context of an IDH mutant glioma, the presence of ATRX nuclear expression is associated with oligodendroglioma but requires confirmatory genetic testing for 1p19q-codeletion (LOH PCR, FISH, copy number microarray, etc.). This is also associated with *TERT* mutation. The presence of nuclear ATRX expression is also seen in a majority of IDH-wildtype glioblastomas, while loss of nuclear staining is associated with IDH-mutant astrocytoma	Can be technically challenging to interpret. Best performed in conjunction with internal positive controls such as endothelial cells.
**H3 p.K28M** (**K27M**)	Nuclear POSITIVITY in tumour cells expressing the mutant H3 p.K28M (K27M) epitope	This mutant epitope is found in DMG and defines this entity. However, there are other less common alterations that are also associated with this diagnosis.	Can be technically challenging to interpret and DMGs harbouring less common disease-defining alterations affecting codon 27 (e.g., H3 p.K28I [K27I]), EZHIP over-expression, or *EGFR* mutation are not identified with this antibody. Note this antibody should be used in conjunction with the H3 p.K28me3 (K27me3) antibody to increase accuracy. This alteration is not specific to DMG and can also be relevant in the workup of ependymomas.
**H3 p.K28me3** (**K27me3**)	Nuclear POSITIVITY in tumour cells is associated with the preservation of the tri-methylated mark of H3 K27, which is anti-correlated with mutations of H3 K27.	LOSS of H3 K27me3 nuclear reactivity supports the POSITIVE IHC finding of H3 K27M mutation. It also provides supporting evidence of a DMG in cases with non-H3 K27M drivers.	Interpretation is best performed in conjunction with H3 p.K28M (K27M) IHC.
**H3 p.G35R** (**G34R**)	Nuclear POSITIVITY in tumour cells expressing the mutant H3 G34R epitope	This mutant epitope is found in DHG and defines this entity. However, the less common alteration, H3 p.G35V (G34V), is also associated with this diagnosis.	Can be technically challenging to interpret and false negative immunoreactivity in H3 G34R mutant DHG cases has been described. Additionally, the less common H3 G34V mutant cases are not identified with this antibody.
**BRAF p.V600E**	Cytoplasmic POSITIVITY in tumours with BRAF p.V600E mutation.	Supports the diagnosis of an epithelioid glioblastoma, ganglioglioma, and other low-grade gliomas but interpretation is context dependent.	The p.V600E mutation is found in multiple types of brain tumours and its interpretation has to be made in an integrative manner.

DHG—diffuse hemispheric glioma; DMG—diffuse midline glioma; IDH—isocitrate dehydrogenase; IHC—immunohistochemistry; FISH—fluorescence in situ hybridization; LOH—loss of heterozygosity; PCR—polymerase chain reaction.

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
