# Peer review of "From Theory to Practice: Implementing the WHO 2021 Classification of Adult Diffuse Gliomas in Neuropathology Diagnosis"

_brainsci, 2023, doi:10.3390/brainsci13050817_

Round 1
Reviewer 1 Report
Succinct and practical review of the application of the updated diagnostic criteria for adult diffuse gliomas. Overall it is comprehensive and well-organized, and reads well. Figure 2 and Table 1 are particularly useful resources to familiarize readers with the diagnostic workup of these tumours.
The manuscript would benefit from a conclusion / concluding statement that summarizes the overall review.
Suggest clarifying the role of CDKN2A in the grading of oligos - the WHO classification does indicate that it may serve as a molecular marker of CNS WHO grade 3 oligo and testing may be helpful for oligos with borderline histological high grade features.
For GBM, the histological description of conventional GBM doesn't include the fibrillary astrocytic nature of the cells. Also suggest further clarifying epithelial metaplasia (vs. epithelioid subtype) given the similarity of these terms. providing a percentage for how many have p53 mutations to better define the commonality of this finding. Consider also mentioning that this is more frequent in giant cell glioblastoma,
A few suggestions for Table 1 for better clarity
- for diagnostic implications for IDH1: add "IDH-mutant" before "astrocytoma vs. oligodendroglioma"
- for diagnostic implications for ATRX: state that the confirmatory genetic testing is for 1p19q codeletion. To the neophyte, the way it is written could imply that the confirmatory testing is for ATRX. Also consider adding a statement as to the diagnostic implications of loss of nuclear staining (in the context of IDH-mutant gliomas), as well as IDH-wildtype gliomas.
- for K27me3, it is worth mentioning that this alteration not specific to diffuse midline gliomas (i.e. also relevant to the workup of ependymomas)
Suggest labelling and explaining what the FISH images are demonstrating in Figures 3 and 4
For Figure 4, the wording could be interpreted to mean that both gliosarcoma and epithelioid GBM frequently harbour BRAF mutations. suggest either making description of F separate sentence or adding "the latter of which... " before "frequently harbours"
References are comprehensive, but appears to be missing a direct reference to the WHO CNS5.
Minor editing needed.
(1) singular-plural mismatches (page 2 line 72; page 7 line 267 [omit "a"]; page 7, line 279; page 9 line 374 - results)
(2) Awkward wording / pronouns
- page 3, line 106, omit "an" and change end of sentence to "are seen"
- page 3, line 108: "...plump endothelial cells, which sometimes form glomeruloid structures.
- page 4, line 143: suggest: "........IDH-mutant (grade 4) astrocytomas and IDH-wildtype glioblastomas clinically behave differently, with IDH-mutant..."
-page 7, line 279, missing "the" before "majority of glioblastomas"
- page 9, line 369, needs a comma between substantial and depending.
- page 9, line 370, needs "a" before next generation sequencing assay
- page 10, table, for G34R, caveats: change "have" to has
(3) Missing dash - page 6, line 248: progression-free survival
Reviewer 2 Report
The manuscript can be accepted after the authors correct the following comments:
1. The title unclearly and insufficiently reflects its content. Please rewrite in a simpler way for the readers.
2. The abstract is written well. The aim of this review article is clear. No further change required for this section.
- Please note that the introduction part needs further paragraphs. Kindly, add new paragraphs at the introduction section with modern references.
4. The resolution of manuscript figures is clear. However, the figures captions are too long and needs to remove the redundant lines.
5. Section 4.3 made of too long paragraph. Could you please separate it to two paragraphs are linked together?
6. Section 5 also consist of one long paragraph. Could you please separate it to two paragraphs are linked together?
- This article lacks the conclusion section. So, please write it to reflect the content in a good way.
- Make sure that all sentences are linked together.

Round 2
Reviewer 2 Report
No further comments.
All the best